# Bioengineering Bone Tissue with 3D Printed Scaffolds in the Presence of Oligostilbenes

**DOI:** 10.3390/ma13204471

**Published:** 2020-10-09

**Authors:** Francesca Posa, Adriana Di Benedetto, Giampietro Ravagnan, Elisabetta Ada Cavalcanti-Adam, Lorenzo Lo Muzio, Gianluca Percoco, Giorgio Mori

**Affiliations:** 1Department of Clinical and Experimental Medicine, University of Foggia, viale Pinto 1, 71122 Foggia, Italy; adriana.dibenedetto@unifg.it (A.D.B.); lorenzo.lomuzio@unifg.it (L.L.M.); giorgio.mori@unifg.it (G.M.); 2Department of Biophysical Chemistry, Heidelberg University and Max Planck Institute for Medical Research, Jahnstraße 29, 69120 Heidelberg, Germany; eacavalcanti@mr.mpg.de; 3Glures srl. Unità Operativa di Napoli, Spin off Accademico dell’Università di Venezia Cà Foscari, Via delle Industrie 19b-30175 Venezia, Italy; ravagnangp@gmail.com; 4Department of Mechanics, Mathematics and Management, Polytechnic University of Bari, Via E. Orabona 4, 70125 Bari, Italy; gianluca.percoco@poliba.it

**Keywords:** bone, mesenchymal stem cells, biomaterial, polycarbonate, resveratrol, polydatin, osteogenic differentiation, focal adhesions, bone health

## Abstract

Diseases determining bone tissue loss have a high impact on people of any age. Bone healing can be improved using a therapeutic approach based on tissue engineering. Scientific research is demonstrating that among bone regeneration techniques, interesting results, in filling of bone lesions and dehiscence have been obtained using adult mesenchymal stem cells (MSCs) integrated with biocompatible scaffolds. The geometry of the scaffold has critical effects on cell adhesion, proliferation and differentiation. Many cytokines and compounds have been demonstrated to be effective in promoting MSCs osteogenic differentiation. Oligostilbenes, such as Resveratrol (Res) and Polydatin (Pol), can increase MSCs osteoblastic features. 3D printing is an excellent technique to create scaffolds customized for the lesion and thus optimized for the patient. In this work we analyze osteoblastic features of adult MSCs integrated with 3D-printed polycarbonate scaffolds differentiated in the presence of oligostilbenes.

## 1. Introduction

The global increase in average life expectancy is leading to an escalation of age-related health problems which may affect organs or tissues. Although the bone tissue is capable of self-regeneration, there are several pathological conditions, determined by serious trauma or degenerative diseases, which require appropriate medical procedures in order to realize a complete recovery of the anatomical and functional properties of the tissue. These innovative therapeutic approaches are part of the so-called regenerative medicine [1,2]. The use of adult stem cells in bone reconstructive therapies offers significant benefits [3,4].

In fact, it is known that stem cells are capable of self-renewal and can differentiate into different cell types, ensuring the repair of most tissues, thus becoming highly useful for tissue engineering. Stem cells in human adults can be isolated from various tissues, including bone marrow, nervous tissue, peripheral blood, retina, liver, pancreas, tooth, and dental bud. In particular, among adult stem cells, mesenchymal stem cells (MSCs), originally identified in the bone marrow, which is still considered the best cell source for osteogenic differentiation, can be isolated also from several adult tissues such as adipose tissue, dental tissues, skin, brain, liver, and fetal tissues [5,6]. MSCs appropriately isolated and induced to differentiation, through integration with biocompatible scaffolds, could represent a valid therapeutic approach for the regeneration of connective tissues as bone and cartilage, healing defects of traumatic, degenerative or neoplastic origin. Unfortunately, several concerns may arise from autologous and allogeneic stem cell transplants which, in their turn, may also not be sufficient to accelerate the healing process in the case of large bone defects [7]. As a consequence, in the last decades an increasing pivotal role has been attributed to tissue-engineered bone grafts developed through a combined effort of engineering, biotechnology and biomaterial science [8]. In this perspective, especially for hard tissue regeneration, one of the most promising approaches is the use of customized scaffolds combined with factors allowing cell proliferation and osteogenic differentiation [9]. Matching tissue engineering with predictive medicine, based on mechanobiological computational models, would optimize healing processes [10]. The fabrication methods for tissue engineering are conventional methods, additive manufacturing techniques and 4D printing [11]. Among these methods, 3D printing is an encouraging technique, easily available to realize personalized scaffolds to be used for tissue regeneration [12].

Dental bud stem cells (DBSCs) are widely recognized as MSCs that can effectively differentiate into osteoblast-like cells [13] becoming a suitable candidate for bone regeneration. DBSCs express the typical mesenchymal stem markers and, as we have shown, their ability to acquire the osteoblastic features and to produce mineralized bone matrix in vitro, which is the crucial event of osteogenic differentiation [14,15], can be positively influenced by several molecules [16,17,18,19]. Polydatin (Pol) deserves a particular mention among the natural molecules capable of inducing DBSCs osteogenic differentiation [19] and opens new horizons to its possible use as a therapeutic agent, as we have exhaustively detailed in our new invention patent (patent n.16999PTIT entitled “Composizioni comprendenti o costituite da Polidatina per uso nel trattamento delle patologie ossee”—“Compositions comprising or consisting of Polydatin to be used in the treatment of bone pathologies”, deposited with application number 102017000079581). Pol, that is a natural precursor of the polyphenolic compound Resveratrol (Res), is a glucoside we find in abundance in fruits and plants [20]. This glucoside shares some of the beneficial biological properties fully demonstrated for Res [21,22,23], but, in comparison, it also presents advantages: higher stability, significant abundance and better oral absorption [24,25,26]. We have previously shown that DBSCs positively responded to Res and Pol treatment increasing their osteogenic potential and, moreover, Pol appeared to be more effective than Res even when used at a very low concentration [19]. To induce bone regeneration, porous scaffolds with appropriate shape, pore size, porosity, degradability, biocompatibility, mechanical properties and desirable cellular responses are required. 3D printing has revealed to be very useful in this field, thanks to the capability to process complex shapes with a wide variety of biocompatible materials such as poly(l-lactic acid) (PLLA), poly(vinyl alcohol) (PVA), poly(lactic-co-glycolic acid) (PLGA) and polycaprolactone (PCL) starting from filaments and pellets [27]. In this study we investigate the combination of Res and Pol treatment and 3D-printed polycarbonate (PC) scaffolds to study the possible effects of this set-up on MSCs commitment into the osteoblastic lineage. The PC has been chosen to have artificial scaffolds with bio-compatible material that have strength and stiffness near to bone tissue [28].

## 2. Materials and Methods

### 2.1. Ethics

The study was conducted in compliance with the Declaration of Helsinki and the International Conference on Harmonization Principles of Good Clinical Practice. The research protocol was approved by the ethical committee, within the project BIADIDENT num. Rep 4159/2018, at the University of Foggia Ospedali Riuniti, and all participants gave informed consent allowing their anonymized information to be used for data analysis.

### 2.2. Scaffold Preparation

The scaffolds were manufactured using an Ultimaker S5, equipped with a AA nozzle diameter equal to 0.4 mm. At the best of author’s knowledge, this printer can be considered as one of the best compromise between quality, price and system flexibility. The filament was the 3 mm 3DXMAX^®^, a high-heat 3D printing filament made using Lexan^®^. The print temperature was set to 285 °C and the bed temperature was kept to 110 °C, print speed 30 mm/s. The bed and nozzle temperature parameters are not inside the interval suggested by the 3D printer supplier, but since manufacturing time is lower than 5 min, the machine is able to complete the workpiece without damages. The different pore sizes were obtained setting on the Cura slicing software the distance between lines equal, respectively, to 0.75 mm for small pores, 0.9 mm for medium pores and 1.15 mm for large pores [29]. Figure 1 shows the Ultimaker 3D printer and samples of the manufactured scaffolds.

### 2.3. Patients and DBSCs Cultures

The dental buds were collected from ten healthy pediatric donors (eight-twelve years) who were subjected to the third molar extraction for orthodontic reasons; each patient’s parents provided a written informed consent. The study was authorized by the Institutional Review Board of the Department of Clinical and Experimental Medicine, University of Foggia. The dental papilla, which corresponds to the internal section of the dental bud, and contains stem cells of mesodermal origin, was cut in small fragments and enzymatically digested. Single-cell suspensions were harvested by filtration, and seeded and expanded in vitro as already reported [30,31,32]. In the experiments aimed to examine Res and Pol effect on cell adhesion during the osteoblastic differentiation process, DBSCs were seeded at a density of 3 × 10^3^/cm^2^ and cultured in an osteogenic medium consisting of α-MEM supplemented with 2% heat inactivated fetal bovine serum (FBS), 10^−8^ M dexamethasone and 50 µg/mL ascorbic acid (Sigma Aldrich, Milan, Italy). DBSCs were maintained in the osteogenic medium supplemented also with 10 mM β-glycerophosphate (Sigma Aldrich, Milan, Italy), for the evaluation of Res and Pol effects on cell adhesion, proliferation, differentiation and examination of their ability in the induction of matrix mineralization in cultures on biomaterials.

### 2.4. Res and Pol Treatment

Res and Pol extracted from *Polygonum cuspidatum* (Japanese Knotweed), according to the procedure defined in the Patent EP1292320B1, were provided by Prof. Ravagnan. Res and Pol were dissolved in ethanol at 100 mM stock solutions [33] and then added to the culture media under low serum conditions (2% FBS) to the final concentration of 0.1 µM for both of them, as detailed in Di Benedetto et al. [19]. In the experiments control cells were not treated with Res or Pol and served as control group (Ctr), treated cells were exposed to Res or Pol (treatment group), that were added to the media at every renewal (every 3 days).

### 2.5. Immunofluorescence

For focal adhesion staining, DBSCs were cultured on glass coverslips for 4 days and then fixed in 4% (w/v) paraformaldehyde (PFA) in PBS. Cells were then washed with PBS and blocked in 1% BSA, 5% normal goat serum in PBS for 20 min, to avoid non-specific protein binding. The following antibodies were used: α_V_β_3_ antibody 1:100 (clone LM609 antibody, cat. MAB1976, MerckMillipore, Merck KGaA, Darmstadt, Germany), followed by fluorescently labeled goat anti-mouse secondary antibody (Alexa Fluor 488, 2 µg/mL, Invitrogen ThermoFisher Scientific, Waltham, MA, USA). Samples were embedded in Mowiol containing 0.1% (*v*/*v*) DAPI for an additional staining of the nucleus. Cells were imaged by a multispectral confocal microscope Leica TCS SP5. The images were adjusted in brightness and color with ImageJ software (Research Services Branch, Image Analysis Software Version 1.52c, NIH, Bethesda, MD, USA).

### 2.6. Alizarin Red Staining (ARS)

DBSCs capacity to produce mineralized matrix nodules when cultured on the scaffolds was determined by performing ARS at 28 days of culture in osteogenic conditions. After removing the culture media, cells were rinsed with PBS, fixed in 10% formalin at RT for 10 min. Then cells were washed again with deionized water and stained using a 1% ARS solution for 10 min at RT. At the end of the incubation period the ARS solution was removed and cells were washed twice with deionized water and air dried. The quantification of ARS in the red stained monolayer was performed by extracting the dye and by reading the optical density (OD) in triplicate at 405 nm.

### 2.7. Statistical Analyses

Statistical analyses were performed by Student’s *t*-test with the GraphPad Prism version 8.0.2 for MacOS software (San Diego, CA, USA). The results were considered statistically significant for *p* < 0.05 (indicated as § *p* < 0.01, * *p* < 0.001).

## 3. Results

### 3.1. Both Res and Pol Treatments Influence Cell Spreading and Focal Adhesion Assembly via α_V_β_3_ Reorganization

To investigate the influence of Res and Pol on cell adhesion and spreading, which determine, as a consequence, DBSCs exhibition of osteoblastic features, the cells were cultured for 4 days on glass coverslips in absence of treatment (Ctr) or in presence of Res or Pol added to the media (Figure 2a–c). Such a short period of time was chosen because of DBSCs predisposition to proliferate and produce various cell layers when left in culture for a few days, a condition that would not have allowed a clear observation of focal adhesions. We examined α_V_β_3_ integrin subcellular distribution by confocal immunofluorescence. This integrin receptor has already been shown to be crucial for the osteogenic differentiation process of MSCs [30] and Vitamin D or the supramolecular aggregate T-LysYal^®^ (T-Lys) can enhance its expression and clusterization leading to the induction of the differentiation process [16,17]. As observable in Figure 1, in Ctr cells α_V_β_3_ integrin clusters were hardly detectable and only few structures were present at the periphery of the cells (Figure 2a). On the other hand, the presence of the molecules in the osteogenic media clearly induced a higher expression and also a reorganization of α_V_β_3_ integrin (Figure 2b,c). In particular, Pol treatment (Figure 2c) induced the strengthening of α_V_β_3_ adhesion sites by forming more elongated and larger peripheral clusters in comparison to cells treated with Res (Figure 2b).

### 3.2. Res and Pol Treatments Prompt DBSCs Proliferation and Mineral Matrix Nodules Deposition on PC Scaffolds Presenting Pores of Medium Dimension (0.9 mm)

We analyzed the proliferation capacity of our cell model on PC scaffolds and their ability to differentiate into osteoblast-like cells producing mineralized matrix. Thus, we previously demonstrated also by FT-IR microscopic analysis that dental stem cells express osteoblastic features [34].

DBSCs were seeded on the biomaterials, which presented pores of medium dimensions (0.9 mm), and cultured in the osteogenic media without any treatment (Ctr) or exposed to Res and Pol treatments for a period of 4 weeks. Although in the first weeks of culture it was particularly difficult to find cells visible enough to be photographed using a phase contrast microscope (data not shown), after 3 weeks of differentiation, as shown in Figure 3a–c, cells appeared numerous and established strong contacts among them. In particular, in the control (Figure 3a), cells seemed to fill the corners of the scaffold pores, leaving a hole without any cell in the center of them. On the other hand, cells treated with Res and Pol (Figure 3b,c) were able to proliferate and interact with each other to cover the scaffold pores almost totally and worked to close them practically in a uniform way. Furthermore, long term cultures of DBSCs showed that the formation of calcium-rich deposits, evaluated by using the ARS after 28 days of osteogenic differentiation, was evident in the control (Figure 3d) and strengthened in the treatments (Figure 3e,f). Interestingly DBSCs capacity of mineralized matrix production was highly promoted when the scaffolds were used in combination with Pol treatment: the ARS quantification shown in the graph (Figure 3g) revealed that the production of mineral matrix nodules was greater in cells treated with 0.1 µM Res compared to the Ctr (19.65%), and remarkably enhanced when cells were exposed to 0.1 µM Pol (37.84%) if compared to the Ctr.

### 3.3. Combined Effect of PC Scaffolds and Polydatin on DBSCs Proliferation and Mineralization

Since we observed a greater osteogenic potential when the treatment with Pol was present, to further explore the effect of this molecule on DBSCs osteoblastic differentiation, we cultured the cells on PC scaffolds presenting pores of two other different dimensions: small (0.75 mm) and large (1.15 mm). DBSCs were maintained in mineralizing conditions and stimulated with Pol for 28 days until the deposition of mineralized matrix. As shown in Figure 4, DBSCs proliferation on small pore scaffolds advanced with the progress of culture time (Figure 4a–d), and Pol treatment induced a substantial increase in the number of cells attached to the pores, an effect which was particularly evident after three weeks of osteogenic differentiation (Figure 4c,d). After 28 days of culture, we evaluated with ARS how Pol stimulation influenced the mineralization capacity of our cell model and we observed that deposition of mineral matrix nodules was significantly higher in cells cultured with Pol, compared to the control (Figure 4e).

Interestingly, cells cultured on scaffolds presenting pores of large dimensions (Figure 5a–g) did not respond as well as those seeded on scaffolds with small pores. A very low number of cells were able to colonize the pores, the proliferation was not increased by the passing of time and the Pol treatment did not show any clear effect. Moreover, the ARS quantification evidenced that there was no significant difference between Ctr and Pol in the mineralization degree (Figure 5g).

## 4. Discussion and Conclusions

It is well known that the tissue engineering market, which was globally worth about $4.7 billion in 2014, is estimated to reach a value close to $5.5 billion by 2022, considering only the US market. Adult stem cell research is today in an advanced phase of trialing and, in some diseases, cells are already part of therapeutic protocols for the treatment of illness and disabilities [35]. The involvement of precision medicine or even customized medicine proposes the personalization of health care with therapies, practices and/or “tailor-made” medical devices for the specific patient to be treated. The availability of optimized scaffolds, with shapes perfectly matching the lesion, would further reduce tissue regeneration times, especially after highly invasive surgical procedures. 3D printing is an excellent approach to design personalized scaffolds [36,37].

A correct regeneration process of hard tissues, as bone and cartilage, needs a biocompatible scaffold able to promote MSCs differentiation and transform a tissue repair in architectural and functional recovery. Bone lesions have multiple possible shapes and dimensions depending on the trauma or on the course of the chronic degenerative process [38].

In the case of MSCs bioengineering, the grafting site shape, its environment, morphology and dimension are basic for cell engraftment and differentiation [39].

Customized scaffolds, made rapidly and efficiently by 3D printers, could easily reproduce the perfect shape for the lesion and correctly create the ideal niche for MSCs engraftment [40], and their osteogenic differentiation would be optimized by the oligostilbenes Res and Pol.

In our study we found that both Res and Pol stimulated MSCs adhesion to the bone matrix protein Osteopontin via α_V_β_3_ integrin and, specifically, Pol treatment prompted a greater reorganization of this integrin in focal adhesion sites. The elongated strings observed by immunofluorescence (Figure 2b,c) represent the classic arrangement of α_V_β_3_ implicated in focal adhesion complexes. We can speculate that the already demonstrated osteogenic effect of Res and Pol on DBSCs [19] could be also related to the reorganization of α_V_β_3_ integrin in focal adhesions. Moreover, as already known, the development of focal complexes on the surface of scaffolds is an essential event to trigger signals that stimulate MSCs proliferation and osteogenic differentiation [41,42]. Considering these two issues, we can state that oligostilbenes can be considered osteoinductive.

Furthermore, when we integrated MSCs on PC scaffolds, we found that both Res and Pol were able to induce the mineral matrix deposition. Gathering our observations, we can establish that the scaffolds were able to support the production of mineralized matrix, which is the final step and the main event of MSCs osteogenic differentiation, and, in addition, the treatment with the molecules object of our study positively assisted the mineralization process. In particular, in agreement with what we have recently demonstrated [19], Pol treatment induced an increase in the mineralization degree that was higher than the one observed in Res treatment.

Moreover, examining the structure of the scaffolds, we studied whether different pore sizes could affect MSCs acquisition of the osteogenic features. Thus we printed PC scaffolds with pores of 0.75, 0.9 and 1.15 mm; MSCs were cultured on them and induced to osteogenic differentiation. We focused on the use of Pol as treatment since we observed, in the initial experiments, that this molecule had a greater effect in the formation of calcium-rich deposits in differentiated MSCs when compared to Res treatment. The observed gathered data led us to conclude that the cell number tended to gradually decrease as the surface micropore was getting larger and subsequently also the mineralization capacity (Figure 3, Figure 4 and Figure 5). We compared the results to detect the ideal pore size for cell proliferation and osteogenic differentiation and we deduced that the dimension of 0.75 mm represented the best size to be created with the 3D printer, among the different pore sizes analyzed; the smaller pores produce the optimal niche for MSCs to promote bone formation.

Thus, in conclusion, in this context we confirmed the osteogenic potential of Pol treatment on MSCs. Then we made a step forward by finding, in the combination of this treatment with PC scaffolds presenting small-sized pores, an optimal strategy to induce the osteogenic differentiation of MSCs and the subsequent deposition of mineralized matrix.

The results of this study suggest that the integration of the scaffolds, opportunely designed by 3D printing with MSCs, could optimize tissue regeneration; moreover Pol could be considered a promising approach to improve bone regeneration encouraging further studies for a deeper understanding of its biological mechanisms.

## Figures and Tables

**Figure 1 materials-13-04471-f001:**
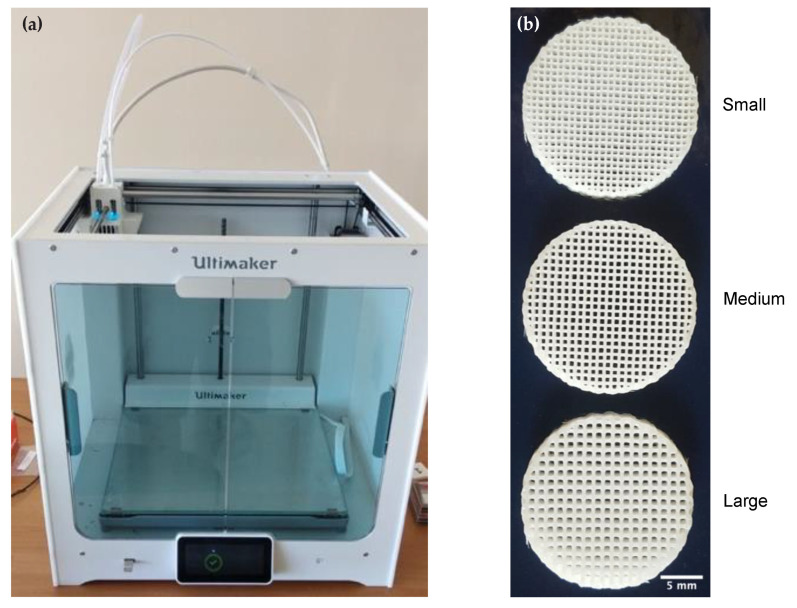
Printer and polycarbonate (PC) scaffolds. (**a**) Ultimaker S5 3D printer. (**b**) Printed scaffolds presenting pores of small (0.75 mm), medium (0.9 mm) and large (1.15 mm) dimensions. Images of representative scaffolds were chosen for the figure. Scale bar = 5 mm.

**Figure 2 materials-13-04471-f002:**
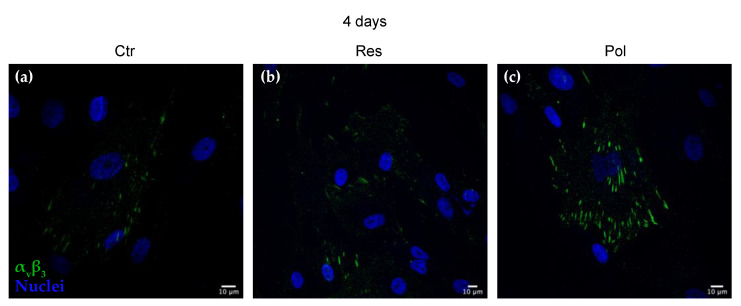
Polydatin (Pol) treatment induces clustering of α_V_β_3_ integrin. Indirect immunofluorescence staining of α_V_β_3_ integrin (green), detected by the antibody LM609, and nuclei (blue) in Dental bud stem cells (DBSCs). Midsection confocal microscopy images show the localization of integrin α_V_β_3_ (green) in cells maintained for 4 days in osteogenic medium and treated with Resveratrol (Res) (**b**), Pol (**c**) and Control (Ctr) (**a**). Images of a representative experiment were chosen for the figure. Scale bar = 10 μm.

**Figure 3 materials-13-04471-f003:**
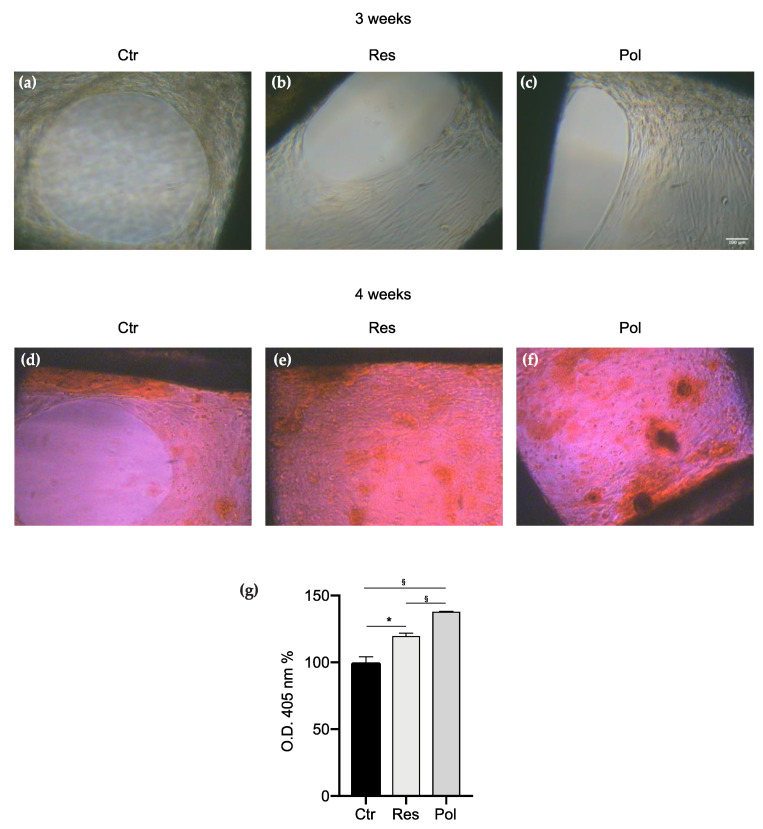
DBSCs proliferation and mineral matrix deposition on medium pore scaffolds. (**a**–**c**) Representative phase contrast pictures of DBSCs treated with Res, Pol or Ctr for 21 days in osteogenic conditions on scaffolds presenting pores of medium dimensions (0.9 mm). Scale bar = 100 μm. (**d**–**f**) ARS (Alizarin red staining) displayed mineral matrix deposition by DBSCs after 28 days of culture. (**g**) The graph shows ARS quantification using the optical density (OD) as mean percentage ± SD and is representative for three independent experiments performed in quadruplicates. § *p* < 0.01, * *p* < 0.001. Student’s *t*-test was used for single comparisons. The biomaterial pores of a representative experiment were chosen for the figure.

**Figure 4 materials-13-04471-f004:**
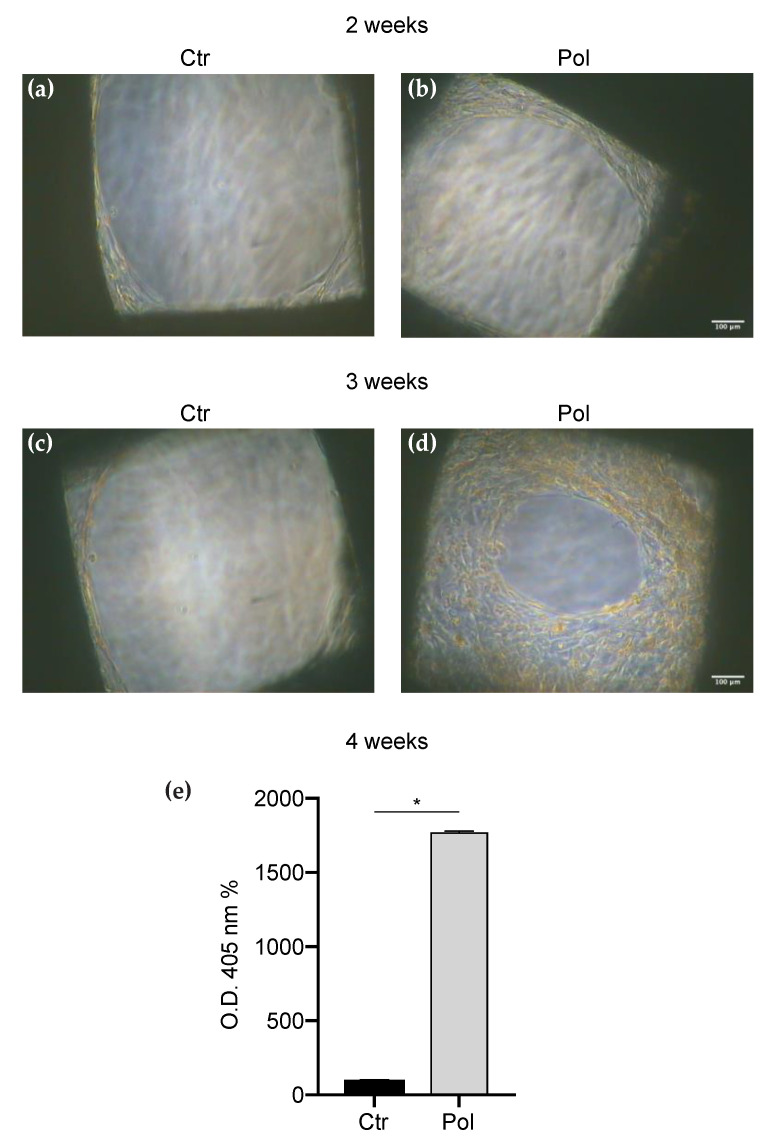
DBSCs proliferation and mineral matrix deposition on small pore scaffolds. (**a**–**d**) Representative phase contrast pictures of DBSCs treated with Pol or not (Ctr) for 14 days (**a**,**b**) and 21 days (**c**,**d**) in osteogenic conditions on scaffolds presenting pores of small dimensions (0.75 mm). Scale bar = 100 μm. (**e**) The graph shows ARS quantification using the OD as mean percentage ± SD and is representative for three independent experiments performed in quadruplicates. * *p* < 0.001. Student’s *t*-test was used for single comparisons. The biomaterial pores of a representative experiment were chosen for the figure.

**Figure 5 materials-13-04471-f005:**
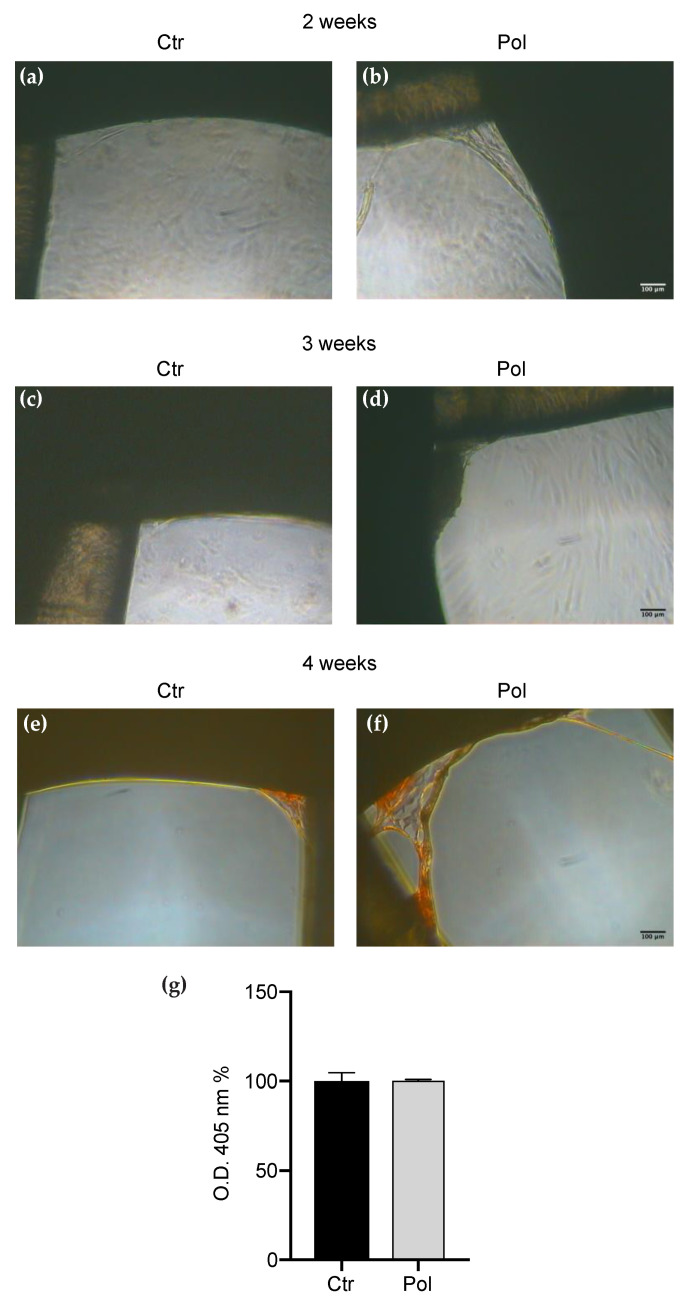
DBSCs proliferation and mineral matrix deposition on large pore scaffolds. (**a**–**d**) Representative phase contrast pictures of DBSCs treated with Pol or Ctr for for 14 days (**a**,**b**) and 21 days (**c**,**d**) in osteogenic conditions on scaffolds presenting pores of large dimensions (1.15 mm). Scale bar = 100 μm. (**e**,**f**) ARS (red staining) displayed mineral matrix deposition by DBSCs after 28 days of culture. (**g**) The graph shows ARS quantification using the OD as mean percentage ± SD and is representative for three independent experiments performed in quadruplicates. Student’s *t*-test was used for single comparisons. The biomaterial pores of a representative experiment were chosen for the figure.

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
