# Peer review of "Bioengineering Bone Tissue with 3D Printed Scaffolds in the Presence of Oligostilbenes"

_materials, 2020, doi:10.3390/ma13204471_

Round 1

Reviewer 1 Report

In this manuscript (Materials-915770), authors have prepared 3D printed scaffolds composed of polycarbonate (PC) and analysed its biological efficacy as bone scaffold in the presence of oligostilbenes. This study could be interesting, but authors have not presented clearly. While gone through the manuscript, I have found some critical points to be considered before acceptance, as follows:

1) Fabrication of 3D printed PC scaffold is described in this manuscript, but Reviewer has not found any digital or morphological image of the 3D printed PCscaffold. Also, Reference is given randomly for this section (Ref 20). Even, in Introduction section, Ref 19 is given for 3D printing of some biomaterials such as PC, but in this Reference no PC is reported for 3D printing purpose. 

2) Authors should provide a detailed fabrication protocol and parameters for 3D printing of PC, including digital images of printer and scaffolds for better and clear information. Also, post-printing difelity and stability (mechanical performance) should be discussed and presented with pore sizes and dimensions.

3) Additionally, this study involves 3D printed scaffold for bioengineering of bone tissue and therefore Introduction section should be eloborated in terms of this data. Authors may refer the following some recent published articles: Front. Mater. 2019, 6: 313. doi: 10.3389/fmats; Applied Materials Today, 2020, 20, 100752; Journal of Manufacturing Processes, 2018, 35, 479-491; etc.

After this major revision, this manuscript can be considered for publication.

Reviewer 2 Report

Congratulations on your study. I enjoyed reviewing it and, clearly, a great deal of work went into completing it.

While I think your study is well done, I believe its presentation can be significantly improved. I would like to categorize my suggestions in a point by point manner.

  1. Conflicts of Interest: You report no conflicts of interest and yet you state that you have a patent application in process for the compound being tested (Patent title [English translation]: “Compounds comprising or constituting polydatin for use in the treatment of bone pathologies”). If readers are going to be able to make an informed decision about how credible this research study is, you must be completely transparent about your potential financial interests. Please adjust the conflicts of interest section accordingly.
  2. The background provided in your abstract should be re-written to better reflect the content of the study. Specifically, the introductory sentence does not seem to make much sense to me in the context of the study.
  3. There are many statements made throughout the article that require citations. For instance lines 58, 243, 248, 257, 260.
  4. At least 40% of the citations you provide are self-citations. This is an area that has seen a great deal of research by many groups in recent years. There are many more recent articles than the ones you cite from your own group that are quite relevant to this topic that you can cite.
  5. In line 39-40, you cite an article from 2013 that speaks of the “potential clinical applications” of hMSCs but you state in the sentence that “therapies offer significant benefits”. If you feel this statement is true, please provide a different citation to justify it or, if not, add the word “potential” to the statement.
  6. In lines 33-34, you discuss to tissues from which MSCs have been harvested but leave out the well-documented largest reserve in the adipose tissue. Please consider adding for completeness.
  7. Consider adding a statement about why, of the various materials available for scaffold production, you chose polycarbonate.
  8. In line 93, please clarify what you mean by “These parameters are to be considered extreme for the Ultimaker”, and if extreme, why you chose to use this particular 3D printer anyway.
  9. Line 113: Although scientifically accurate to use only the description Polygonum Cuspidatum, you may consider adding in parenthesis (Japanese Knotweed) to aid in the ease of reading and understanding.
  10. Line 135 is confusing: “Once discarded the ARS solution…”. Please clarify.
  11. At multiple points in the Results section, you venture off into Discussion and interpretation of your results. These would best be deferred to the Discussion section to make the paper more readable and understandable. Examples: Lines 160-2, 172-3, 237-8.
  12. In line 154, you report that integrin clusters were “hardly detectable”. Was there any objective quantification of this or was is just from viewing under the microscope?
  13. In line 157-8, you report that Pol treatment “induced the strengthening” of adhesion sites. Is there mechanical data to support this assertion?
  14. In line 188, you erroneously refer to Figure 2f, it should be Figure 2g.
  15. Figure 2 a-f are difficult to understand. The understanding of this figure would greatly benefit from labeling and arrows to indicate what you are referring to in the legend.  Additionally, including a photomicrograph or line drawing of the pretreated empty scaffold would greatly aid in the understanding of what you are trying to illustrate. As it stands, I cannot see how it aids in the support of your findings.
  16. In line 222, the phrase “the same considerations cannot be done” is confusing and should be clarified.
  17. In Figure 4, the scale bars should be more readable by changing the color or contrast or enlarging them to a readable font size.
  18. Lines 237-8: You have too few data points to come to this conclusion. You can say there was a trend, however.
  19. Line 278: Your data does not provide enough evidence to claim that 0.75mm diameter pore sizes are the “perfect” size.

Thank you for allowing me the privilege of reviewing your scientific work for publication. I hope this is helpful in improving your paper.

Reviewer 3 Report

The study described in an interesting way.

However, determination of effectiveness in promoting MSCs osteogenic differentiation based on only cell adhesion and spreading is not enough.

Other osteoblastic features of MSCs cells should be assessed, e.g. expression of specific surface antigens.

Minor remark: the abbreviation should be used once (during  its first using) – see: ARS or OD

Round 2

Reviewer 1 Report

Authors have responded well against some points. As bioactivity (i.e. biomineralization) is an important factor in bone tissue regeneration, therefore, authors should describe this point also in Introduction section by including some recent references, such as Front. Mater. 2019, 6: 313. doi: 10.3389/fmats; Journal of Biomedical Materials Research Part A100(10), 2654-2667, etc. After this revision, this manuscript can be accepted for publication.

Reviewer 2 Report

Thank you for addressing the points I had outlined. I think the manuscript presents a much more compelling and focused report of your experimentation. 

In line 62 I believe you mean to say 3D not 4D

Reviewer 3 Report

I have no more comments.

Author Response

I have no more comments.

We thank you for your cooperation.
